# Incorporating the speciation process into species delimitation

**Jeet Sukumaran** [1]*, **Mark T. Holder** [2], **L. Lacey Knowles** [3]

**1** Department of Biology, San Diego State University, San Diego, California, United States of America,
**2** Department of Ecology and Evolutionary Biology, University of Kansas, Lawrence, Kansas, United States of
America, **3** Department of Ecology and Evolutionary Biology, University of Michigan, Ann Arbor, Michigan,
United States of America

☯ These authors contributed equally to this work.
* jsukumaran@sdsu.edu

pcbi.1008924

London, UNITED KINGDOM

**Data Availability Statement:** Data are all
simulated, with simulation methods documented in
paper and scripts available in both Supplemental
Information and on GitHub: https://github.com/
jeetsukumaran/delineate-performance-analysis.git.

## Abstract

The "multispecies" coalescent (MSC) model that underlies many genomic species-delimitation approaches is problematic because it does not distinguish between genetic structure associated with species versus that of populations within species. Consequently, as both the genomic and spatial resolution of data increases, a proliferation of artifactual species results as within-species population lineages, detected due to restrictions in gene flow, are identified as distinct species. The toll of this extends beyond systematic studies, getting magnified across the many disciplines that rely upon an accurate framework of identified species. Here we present the first of a new class of approaches that addresses this issue by incorporating an extended speciation process for species delimitation. We model the formation of population lineages and their subsequent development into independent species as separate processes and provide for a way to incorporate current understanding of the species boundaries in the system through specification of species identities of a subset of population lineages. As a result, species boundaries and within-species lineages boundaries can be discriminated across the entire system, and species identities can be assigned to the remaining lineages of unknown affinities with quantified probabilities. In addition to the identification of species units in nature, the primary goal of species delimitation, the incorporation of a speciation model also allows us insights into the links between population and species-level processes. By explicitly accounting for restrictions in gene flow not only between, but also within, species, we also address the limits of genetic data for delimiting species. Specifically, while genetic data alone is not sufficient for accurate delimitation, when considered in conjunction with other information we are able to not only learn about species boundaries, but also about the tempo of the speciation process itself.

## Author summary

Current coalescent-based species-delimitation approaches rely on the diagnosis of genetic structure to identify putative taxa. However, when multiple population lineages from the same species are sampled, the conflation of populations with species leads to a

**Funding:** LLK and JS received funding from National Science Foundation, Division of Environmental Biology (NSF-DEB) grant 1655607, while MTH received funding from National Science Foundation, Division of Environmental Biology (NSF-DEB) grant 1457776. This funding was critical to the success of this work. The funders had no role in study design, data collection and analysis, decision to publish, or preparation of the manuscript.

**Competing interests:** The authors have declared that no competing interests exist.

proliferation of artifactual "species", resulting in inaccurate diversity estimates that are challenging systematic studies and fields that rely upon accurately delimited species boundaries. We present here a new approach to delimitation that explicitly models speciation as an extended process, from the formation of new population lineages to the development of independent species. This allows for computational discrimination between genetic structure that corresponds to species lineages versus population lineages within species, transforming species delimitation in both theory and practice in this age of high-resolution genomic data.

This is a *PLOS Computational Biology* Methods paper.

## Introduction

Computational (or statistical) species delimitation—the identification or demarcation of species units in nature using algorithmic approaches—is being transformed by unprecedented amounts of genetic data coupled with ever-increasing computational power to process that data. This transformation has relied heavily on the multispecies coalescent (MSC) model (i.e., the censored coalescent model, as originally described [1]). The MSC provides a probability distribution for gene tree shapes from parameters that describe population sizes and the history of divergence times between multiple lineages. In its application in species delimitation, the distinct lineages identified by this model are each equated with being distinct species. However, in systems where there is within-species structure (as in, for example, population lineages), the MSC is problematic for species delimitation [2]. This is because the MSC cannot distinguish between genetic lineages associated with species boundaries from those associated with population divergence *within* species. That is, the MSC detects genetic structure, not species per se [2].

Note that this problem is not an issue of correct or incorrect species concepts, nor is it the result of adhering to any particular view or special model of the speciation process. Rather, *regardless* of the species concept assumed or speciation model adopted by the investigator, whenever detectable genetic structure arises from restrictions in gene flow *before* any speciation, including, notably, population isolation, the MSC will incorrectly and artifactually elevate these population lineages as distinct species. Thus, rather than being just a curious theoretical problem that arises under a peculiar speciation model or particular species concept, for any data in which there is detectable population genetic structure within species, such restrictions in gene flow represent a fundamental and *general* issue with using the MSC to delimit species. Of course, if an investigator considers that all and any restriction of gene flow, however small, as the exclusive and unconditional criteria of defining species boundaries for their particular system, then a pure MSC delimitation analysis will indeed yield results that are consistent with this view. However, in most other cases, and in particular, when in a given system there may be any degree of detectable restriction in gene flow *within* species, however partial or incomplete, then the MSC will still detect these within-species lineages (populations) as distinct units, and when used in a species-delimitation context these will be interpreted as full species by the investigator. This conflation of populations with species, resulting in oversplitting, has been reported many times by many empirical systematists working in a range of systems [3–19]. The recent explicit statistical demonstration and characterization of this phenomenon [2]

simply provided theoretical support for the escalating concern about artifactual species being inferred from coalescent-based species delimitation methods without any corroborating data [18, 20, 21].

Recognizing the limitations of MSC-based applications for species delimitaton, some researchers have proposed a return to heuristic approaches [22, 23]. For example, genealogical indices developed more than a decade ago [24] might be applied in more elaborate statistical frameworks to evaluate species status. Specifically, parameters of population divergence (namely, $\theta$, $\tau$, and $M$) estimated under the MSC with either a summary method [22] or full-likelihood method [23] can be used to calculate a genealogical divergence index, *gdi*. Species status is then determined by a threshold value (e.g., *gdi* > 0.7; [22, 23]), which was based on analyses of a few particular empirical datasets and not based on, or otherwise informed by biological or statistical theory.

Heuristics are not the answer to addressing inadequacies of the MSC for species delimitation. In fact, heuristic criteria for interpreting the results from the MSC are less than ideal, just as they were when they were originally proposed decades ago. Heuristic criteria may be disconnected from speciation: for example, while monophyly criteria can easily be applied to identify species boundaries, monophyly across the genome is not reached until many generations after species divergence [25]. Sensitivity of heuristic criteria to processes unrelated to speciation can also give rise to misleading interpretations. For example, if a population is founded by a few individuals, or if the two populations have very different sizes, indices become unreliable for making interpretations about species status (e.g., elevated values of *gdi* result because of the dependency of index on population divergence time relative to population size; [23]).

Here we introduce a new framework—speciation-based delimitation—that distinguishes species and population boundaries using full probabilistic models. Specifically, by incorporating an explicit model of an extended speciation process into the delimitation analysis (Fig 1), the formation of population lineages and their subsequent development into independent species are decoupled and modeled separately. Furthermore, this framework allows for the incorporation of existing systematic information in the form of known species identities for a subset of population lineages in the study. This information will be used to estimate a tempo of speciation, which can then be used to estimate more reliable and accurate species delimitations for the remaining subset of population lineages of unknown or uncertain species identities. In this way, our approach also implicitly captures the species concept used by the investigator or considered by the investigator to be appropriate for the particular system, rather than enforcing any particular pre-defined species concept on the analysis. In the following sections, we describe in detail the approach implemented in the new software DELINEATE, followed by presentation of two different types of delimitation analysis, as well as a novel macroevolutionary analysis of diversity. We then assess the performance of our approach using simulations under a broad range of conditions that include challenging parts of parameter space to fully characterize the approach's strengths, as well as its weaknesses. What we present is certainly not the only, or necessarily the best, speciation model that might be incorporated into delimitation analyses. Nonetheless, our work represents a significant step towards a future where not only the full potential of genetic data can be realized through model-based inference, but the limits of genetic data for delimitation are also explicitly addressed.

## Materials and methods

### Types of inferences under DELINEATE

DELINEATE has three modes of inference, depending upon the goals of a study, the ancillary information a researcher has on the focal taxa, and/or the study design. For example, in most

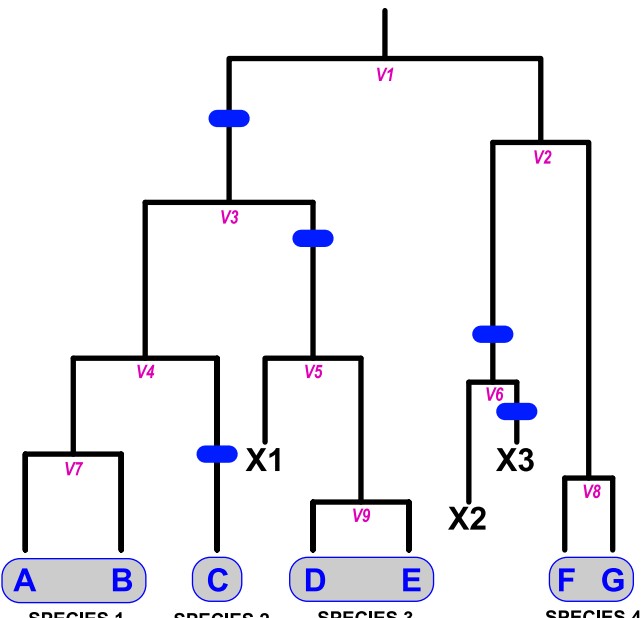

**Fig 1. The Protracted Birth Death (PBD) model of the speciation process [26, 27] implemented in `DELINEATE` separately models lineage splitting and completion of speciation events, such that "speciation" is an extended process.** For example, considering a phylogeny of population lineages inferred under the MSC [1, 28], the lineage splitting events correspond to the formation of new isolated *population* lineages (not species) through restrictions in gene flow in an ancestral population (e.g., $V_1$). These lineages may themselves give rise to other population lineages ($V2$ through $V9$), or go extinct ($X1$ through $X3$). Population lineages develop into an independent species at a fixed background rate, providing they are not otherwise lost (i.e., there is duration between the initiation and completion of speciation). Changes in status from incipient to full or good species are marked by speciation completion events, shown by the blue bars. Under the PBD, a "species" is thus made up of one or more population lineages not separated from one another by a speciation event. In this example, five speciation completion events divide the seven extant populations into four species: {*A*, *B*}, {*C*}, {*D*, *E*}, and {*F*, *G*}.

species delimitation applications, we very rarely have absolutely *no* knowledge of any species identities of the sampled individuals. Instead, we typically have some understanding of the species assignments of some of the individuals in a dataset, often by design, and it is the species identities of only a subset of the collected data that we are interested in actually inferring. In such cases, the "constrained" mode of inference would be applied in `DELINEATE`, rather than the "unconstrained" inference mode. Alternatively, the "tempo of speciation" mode of inference might be preferred if the focus of the study is on speciation dynamics, rather than delimitaion per se (see below).

**Constrained species delimitation.** Under this mode of inference, the species identities of *some* of the lineages in the input tree are specified *a priori*. `DELINEATE` will assign species identities to the remaining lineages of unknown species identities, with these estimated species assignments being either to existing (i.e., known or specified) species, or new ones entirely. In particular, with species identities of a *subset* of lineages specified as constraints on possible partitions to be evaluated, `DELINEATE` will estimate the speciation dynamic parameter (specifically, the speciation-completion rate, $\sigma$) based on the tree induced by this subset, and then calculate the probabilities of all possible partitions that include the species identities given the estimated speciation-completion rate. The different partitions can then be ranked according to their probabilities, with the partition of the highest probability constituting the maximum likelihood estimate of the delimited species boundaries. (Note that the speciation-completion rate

or $\sigma$ is the primary parameter that regulates the number of species in the system and captures the tempo of species formation as opposed to population lineage isolation or fragmentation. It is the rate that an independent population lineage develops full distinct species status. This is described in detailed in the "Statistical Model Description and Inference Algorithm" below.)

In addition to identifying the most probable species assignments while estimating parameters of the speciation process (i.e., $\sigma$ is unknown), this mode of inference in DELINEATE could also be used to focus upon the identities a particular subset of lineages (e.g., a set of conspecific lineages for a particular species). By integrating across partitions (i.e., alternative species assignments) and $\sigma$, the probability of their identities—whether or not they belong to the same species or not—can be summarized taking into account uncertainty across delimitation models (i.e., uncertainty in the identities of non-target taxa, as well as the speciation dynamics).

We acknowledge that in the context of estimating $\sigma$ from genetic data and the species constraints, the lineage tree is not independent of the species constraints. That is, each heterospecific constraint is more compatible with a longer path along the lineage tree, and each conspecific constraint is more compatible with shorter paths. Given that the information about path length from the species constraints is likely to be much weaker than the information provided by the genetic data, here we approximate the results by performing sequential inference of the lineage tree and the $\sigma$ parameter value. The dependence would require joint inference of the lineage tree and $\sigma$, which is beyond the scope of this paper. The performance measures as reported here show generally high accuracy, and that accuracy is primarily affected by other factors (e.g., number of lineages and the tempo of speciation).

Using the "constrained" mode of DELINEATE provides a straightforward and elegant way to incorporate the knowledge, insight, expertise, and perspective that investigators have about a particular system into the species delimitation analysis. This would be intuitively (though not statistically or operationally) comparable to the prior in a Bayesian analysis or a training dataset in a machine learning analysis. This information is communicated to the DELINEATE analysis through the constraints, i.e. the assignment of species identities to a subset of the population lineages. As discussed above, DELINEATE uses information from the assignment of species identities specified in the constraints to learn about the speciation process (i.e., the speciation completion rate) and estimate the probabilities of the species identities of the unknown population lineages. As such, these estimates will reflect the investigator's understanding of "species; for the data being analyzed. That is, different species concepts are accommodated (and reflected) in the constraints specified by the investigator, and the probabilistic assignment of unknown lineages to new or existing species will be consistent with concept provided by the investigator.

**Unconstrained species delimitation.**   The most probable species assignments may be inferred without specifying any information about known or believed species identities with DELINEATE, an analysis type hereafter referred to as unconstrained species delimitation. However, this mode of inference requires the input of speciation dynamic parameters (in our model, the speciation-completion rate). Information for setting a given $\sigma$ in empirical studies might come from estimates of $\sigma$ from a comparable (but distinct) data set, such as a related group of species for which information on species identities can be used to estimate $\sigma$ (e.g., using DELINEATE in the constrained species delimitation mode, as already described), and for which similar speciation dynamics might be assumed.

**Tempo of speciation dynamics.**   An interesting and useful application of DELINEATE is to study the temporal dynamics of speciation. In particular, with estimates of the speciation-completion rate, we can calculate the *speciation-completion time*, that is, the waiting time for a *single* isolated population or incipient species lineage to complete the speciation process

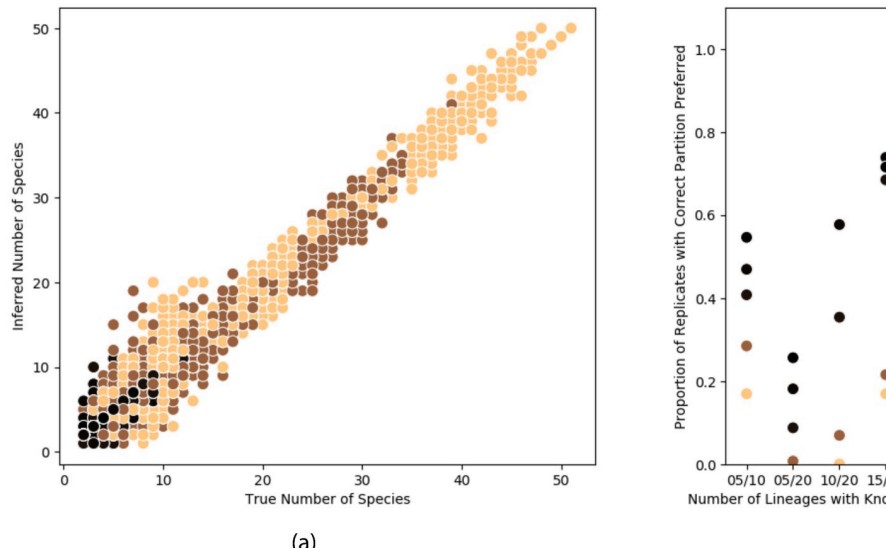

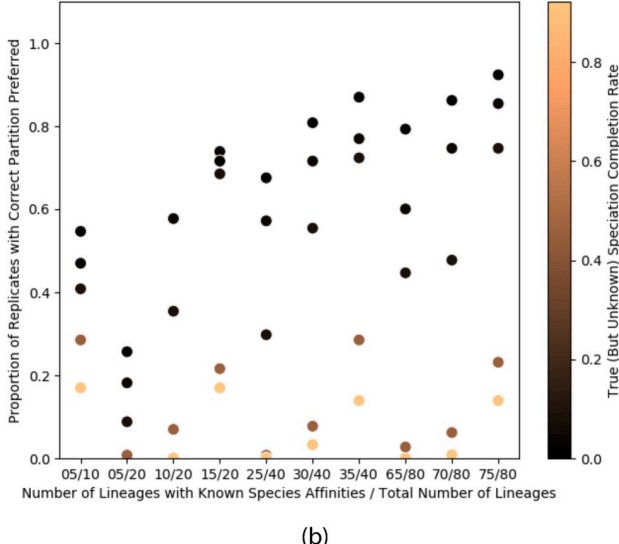

(a)

(b)

**Fig 2. Overview of speciation-based delimitation using `DELINEATE`.** Starting with genomic data (a), a lineage tree (b) is inferred under the multispecies coalescent (MSC) model using any of a number of programs, such as BP&P [28] or *BEAST [30]. The inferred lineages, which are consistent with a Wright-Fisher population under the MSC model (i.e., cannot be divided any further), are (c) organized into sets of one or more species in `DELINEATE`, with each possible organization (referred to as a "partition") representing a different hypothesis about species boundaries. Partitions can range from a single species (i.e. all lineages assigned to the same species) to as many species as their are lineages (i.e., there are no population lineages, only different species). The probability of each of the different partitions is calculated and reported by `DELINEATE`, The partition with the highest probability is the maximum likelihood estimate, but investigators have at their disposable all the partitions in the 95% confidence interval as well if they wish to summarize support for particular results as well.

assuming it does not go extinct. Under the PBD model, there is no known analytical solution or maximum likelihood estimator for the speciation-completion rate, $\sigma$ [29]. However, `DELINEATE` can provide an estimate of the speciation-completion rate, $\sigma$, either by optimizing the parameter during the course of a standard constrained species delimitation analysis (see above), or by running `DELINEATE` in a special mode where the species assignments of all lineages are given and $\sigma$ is the only unknown parameter estimated.

## Statistical model description and inference algorithm

We treat the data as a set of samples of sequences (***D***) for $K$ loci from $M$ populations (lineages), with $N_m$ individuals sampled from population $m$. A population tree with branch lengths is inferred from these sequences using the "censored coalescent" or multispecies coalescent (MSC) model of [28]. In our current implementation or `DELINEATE`, the inference of the population tree under the MSC thus represents the first stage of analysis (Fig 2). Following the notation of [28], $\Lambda$ represents the assignment of populations to species, and $S$ is tree of population relationships, with branch lengths, estimated from multilocus data resulting from inference under the MSC model. The posterior probability of a particular partition given the data would require taking an integral over all possible trees of populations and all possible values of the other parameters of the model. While this could be approximated using Markov chain Monte Carlo methods, here we assume that species assignment is independent of the process of molecular evolution, once we condition on the lineage tree. Our approach focuses on calculating the likelihood of a species partition given a lineage tree and the speciation-completion

rate, $\sigma$:

$$\mathcal{L} = \Pr(\Lambda \mid \sigma, S)$$

To take uncertainty in the lineage tree into account, one could sample lineage trees from their posterior distribution (e.g., generated using software such as *BEAST), and average $\Pr(\Lambda \mid \sigma, S)$ over this sample.

Information about the speciation-completion rate is inferrable from the lineage tree, $S_\mathcal{C}$, induced by the set of taxa the researcher has previously classified as the constrained leaf set ($\mathcal{C}$); we denote the species partition of this subset of leaves as $\Lambda_\mathcal{C}$. We assume that $\mathcal{C}$ is a random sample of the full leafset to calculate $\Pr(\Lambda_\mathcal{C} \mid \sigma, S_\mathcal{C})$ as the likelihood for $\sigma$, otherwise, some sort of correction would be needed for an ascertainment bias.

We use a Poisson distribution to obtain the probability of 0 or $\geq 1$ speciation completion events occurring along a branch given an instantaneous speciation-completion rate of $\sigma$. That is, given a branch with duration time of $\tau_i$, then the probability of the speciation process *not* completing on this branch is given by:

$$\Pr(\text{No speciation completion events} \mid \sigma, \tau_i) = e^{-\sigma\tau_i},$$

while the probability of the speciation process completing is given by:

$$
\begin{aligned}
\Pr(\text{Speciation completion} \mid \sigma, \tau_i) &= \Pr(\text{One or more speciation completion events} \mid \sigma, \tau_i)\\
&= 1 - \Pr(\text{No speciation completion events}\\
&= 1 - e^{-\sigma\tau_i}.
\end{aligned}
$$

Note, thus, that the conspecific/heterospecific status of an ancestor-descendant pair only depends on whether the number of speciation completion events is 0 or greater than 0. Despite only having to keep track of two states for each branch (0 vs > 0 speciation completion events), the number of possible configurations over the entire lineage tree ($S_\mathcal{C}$) can be quite large: $2^{2|\mathcal{C}|-2}$, where $2|\mathcal{C}| - 2$ is the number of branches in a rooted tree. Fortunately, calculating the likelihood, $\mathcal{L} = \Pr(\Lambda_\mathcal{C} \mid \sigma, S_\mathcal{C})$ only entails summing over those configurations of speciation completion events across branches that are compatible with all of the constraints implied by $\Lambda_\mathcal{C}$. For a fixed lineage tree and a moderate size of $\mathcal{C}$, this calculation is feasible via dynamic programming. Bookkeeping similar to Felsenstein's pruning algorithm can be performed during a post-order traversal of the tree to determine the likelihood of the constrained leaf partition given a tree and $\sigma$ (details of the dynamic programming algorithm used are provided in S1 Text.).

Note that if the researcher's prior information on alpha taxonomy only consists of conspecific constraints, then $\hat{\sigma} = 0$ because in that case the probability of all lineages belonging to the same species would have a probability of 1. Similarly, if the researcher's constraints are all heterospecific constraints, then $\hat{\sigma} = \infty$ because this will assign a probability of 1 to a scenario in which every tip is its own species. Thus, one would expect a plausible estimate of $\sigma$ only when the set of input constraints has at least one conspecific and at least one heterospecific constraint.

## Performance assessments

The performance of each of the different modes of inference available within our full probabilistic model for species delimitation was evaluated using simulated data. These assessment include identifying the limitations of DELINEATE in addressing the various delimitation objectives discussed above. In all cases, post-inferential statistical analyses and visualization were done using the *pandas* [31], *seaborn* [32], and *Matplotlib* [33] Python libraries. The 95%

confidence intervals were calculated by adding partitions in decreasing order of probability until 95% of the (constrained) probability is accounted for (i.e., the interval consists of the set of the highest-probability partitions, with the sum of probabilties equal to at least 0.95).

**Constrained species delimitation mode of inference.** Recovery of the true species partition given *a priori* species assignments for some lineages and an unknown speciation-completion rate (i.e., *joint* estimation of the species partition and $\sigma$) was assessed by generating 100 replicate datasets across a range of parameter values and dataset sizes, and for varying proportions of unknown species identities per dataset randomly assigned to lineages. Specifically, we simulated datasets with each distinct combination of the following parameters:

- number of lineages (i.e., tips in tree): 20, 40, and 80,

- number of unconstrained lineages (i.e., lineages of unknown species identities) that may either belong to an existing or new species: 5, 10, and 15, and

- speciation-completion rate: 0.001, 0.005, 0.010, 0.050, and 0.100.

Each dataset was analyzed where known species identities for the subset of taxa were specified (according to parameters described above); the speciation-completion rate was estimated, not specified. For each analysis in `DELINEATE` we recorded: (a) whether the species partition with the highest probability corresponded to the known or true species partition, and (b) whether the true species partition was in the 95% credibility set.

**Unconstrained species delimitation mode of inference.** To assess the accuracy of the species partition probability calculation when *no* species constraint information is provided, we generated test data by simulating species on a random population tree under a known speciation-completion rate using the "`ProtractedSpeciationModel`" class in the DendroPy phylogenetic computational library. The true speciation-completion rates varied from 0.01 to 0.1 in 0.01 increments, and population isolation and extinction rates were fixed to 0.1 and 0.0, respectively. Due to computational limits with enumerating all possible species partitions, the population tree size was limited to 15 tips, and only 10 replicates were conducted under each speciation-completion rate.

**The tempo of speciation mode of inference.** To assess the accuracy of the estimates of the speciation-completion rate, we first generated test data by simulating species on a random population tree with a known speciation-completion rate. More specifically, using the "`ProtractedSpeciationModel`" class in the DendroPy phylogenetic computational library, we generated 40 tip population trees with fixed population isolation rate of 0.1, extinction rate of 0.0, and the following speciation-completion rates ($\sigma$): 0.001, 0.002, 0.004, 0.008, 0.01, 0.02, 0.04, 0.08, and 0.1. Thus, across all cases, speciation-completion rates varied from 100 times slower to equal to the population isolation rate. This range of values, from 0.01 to 1.0 relative to the population isolation rate, spans (and exceeds) the relative range of speciation-completion rates reported for a variety of empirical systems by [27]. A total of 100 replicates were run under each configuration, and the resulting trees and associated true species partitions were each submitted as data to the tempo of speciation mode of inference in `DELINEATE` to calculate the maximum likelihood speciation-completion rate, which was compared to the known (true) value used to simulate the data. 95% confidence intervals were calculated using the Fisher information approach [34].

## Software and data

Software for inference under the `DELINEATE` model is publically available at: https://github. com/jeetsukumaran/delineate This software is written in Python [35], and makes use of the

*NumPy* [36], *SciPy* [37], and *DendroPy* [38] software libraries. The package is fully documented, with documentation available online at https://jeetsukumaran.github.io/delineate/. The documentation includes a primer on background concepts as well as a fully worked empirical case example using a *Lionepha* dataset recently published by [39], illustrating analytical procedures and guides.

Scripts that were used generate data and analyze them for the performance tests are available S1 Data.

## Results

By simulating across a broad range of parameter space, we identify properties of delimitation analyses that can be accurately inferred and are generally robust to different study conditions, as well as those whose accuracy varies, thereby informing which modes of inference in DELINEATE might be more or less appropriate for a specific study. With respect to inferring the number of species [Fig 3a], DELINEATE performs very well using the constraint mode of inference. Specifically, regardless of data set size and the number of lineages with inferred (as

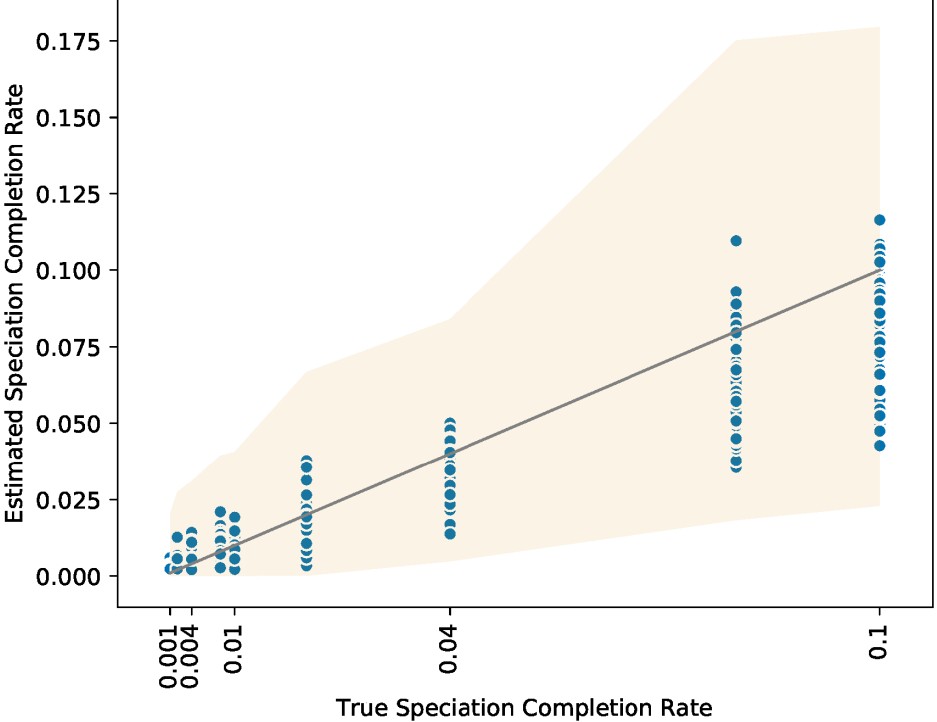

**Fig 3. Accuracy of species delimitation under different levels of species constraints and dataset sizes (i.e., number of lineages in the tree).** Simulations span differing speciation-completion rates (indicated by color gradient with darker colors representing lower rates and lighter colors representing higher rates). Even with inferring the speciation-completion rate from the data, (a) recovery of the correct number of species is extremely reliable across a broad range of conditions, comparing the true number of species with the the inferred number of species; each dot corresponds to the analysis of one replicate dataset. However, whether the (b) identity of species is accurately inferred differs depending upon the size of the data set (i.e., number of lineages), the constraint level (i.e., the number of lineages with designations set a priori; e.g., "30/40" corresponds to a tree with 40 lineages, 30 of those with known species identities, and 10 lineages with inferred identities), and the particular speciation-completion rate the data were simulated under (note that this rate was inferred during the analyses). Shown are the proportion of 100 replicates for each set of conditions in which the partition with the highest probability corresponded to the correct assignments of all species identities.

                                    

opposed to *a priori* assigned) species identities, the estimated number of species shows a strong correspondence with the actual number of species in the simulated data sets. Moreover, estimates of the number of species were robust to differences in the underlying speciation dynamics (i.e., were insensitive to the speciation-completion rate, which was also estimated).

The accuracy of inferred species identities, unlike estimates of the number of species, varied depending on the speciation dynamics and properties of the dataset [Fig 3b]. With respect to correctly assigning *all* lineages to their respective species designations, three trends are apparent from the analyses. Unsurprisingly, the more constraint information provided in terms of known species identities of lineages, the better the performance. The proportion of replicate datasets with the correct inferred species identities across all lineages also increases with dataset size (i.e., with increasing numbers of lineages in the tree), and with lower speciation-completion rates. Despite the obvious sensitivities to these conditions, when inferring the species identity of 5 out of the 20 lineages under a speciation-completion rate of 0.001 (i.e., one hundred times lower than the population isolation or splitting rate of 0.1), the partition with the highest maximum likelihood corresponded to the correct species identities for all lineages in almost 80% of the replicates. For larger datasets of 80 lineages, the identity of 5 unknown lineages is correctly inferred in 96% of the replicates for the same speciation-completion rate [Fig 3b]. Also note that in all these analyses under the constrained mode of inference, the speciation-completion rate is also inferred, thus we learn about the tempo of speciation in addition to the species boundaries.

Note that the analyses span a range of dataset configurations with respect to the number of species, or conversely population, lineages (see the range of values for any single speciation-completion rate in [Fig 3a]). However, regardless of dataset size or number of constraints (i.e, the number of lineages with an assigned species designation), the correct delimitation model was rarely inferred under high speciation-completion rates—that is, as the speciation-completion rate is on the same order or as much as half the population isolation rate. Such rates are not likely to be biologically realistic; they are presented here to illustrate changes in performance across the entire theoretical parameter space, not to represent rates that are likely to be apply in practice.

For data analyzed using the unconstrained mode of inference, which was restricted to smaller dataset sizes because of computational constraints (see Materials and methods for details), 925 out of a total of 1000 replicates across all speciation-completion rates (i.e., in approximately 92.5% of the replicates) the true species partition was recovered in the 95% confidence interval. For these analyses of 15 lineages per replicate, the true species delimitation model (i.e., partition) ranged from 2 to 12 species, with an average of approximately 5 distinct species per replicate (mean = 5.092; s.d. = 2.1217). However, the number of partitions in the confidence intervals of the DELINEATE analysis ranged from 108 to 187,554, with a median of 16,832 and a mean of 23,625. Moreover, while the true species partition was in the confidence interval most of the time, only in 12 cases (i.e., 1.2%) was the true species partition (i.e., the delimitation model with all lineages assigned correctly) the one with the highest probability. Overall, the results demonstrate that while we are able to reduce the uncertainty of species assignments, the residual level of uncertainty indicates that genomic data alone is insufficient to infer all species identities. This is because, as pointed out by [23], the speciation process is conditionally independent of the population tree.

Tests to evaluate the performance of DELINEATE for estimating the speciation-completion rate showed that the maximum likelihood estimates generally tracked the true speciation-completion rates well (Fig 4). The estimates were particularly good at low to moderate speciation-completion rates, and tended to be underestimated when the true speciation-completion rate was high. This is most likely due to saturation, where, at high rates of speciation-completion,

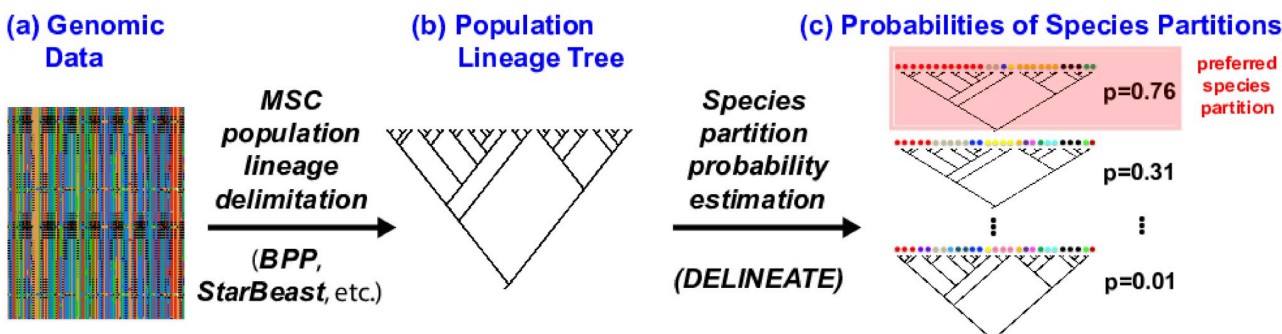

**Fig 4. Accuracy of `DELINEATE` maximum likelihood estimates of speciation-completion rate compared with the true speciation-completion rate; the ribbon shows the minimum and maximum of the 95% confidence interval ranges for the various estimates under that true rate.** Note that the population isolation rate was fixed at 0.1, so the range of speciation-completion rates, from 0.001 to 0.1 spans rates from one hundred times slower to equal to the population isolation rate.

multiple speciation completion events may occur along the same branch without changing the shape of the data in terms of distinct species identities at the tips. For the 60 lineage trees analyzed, each replicate ranged from 4 to 37 species each, with an average of approximately 20 distinct species per replicate (mean = 19.97; s.d. = 7.97).

## Discussion

With the new class of speciation-based delimitation we introduce here, we can confidently infer species identities within a reasonable part of realistic parameter space, distinguishing genetic structure within species from that associated with species boundaries, thereby avoiding the overestimation that occurs with applications based on the MSC [2]. Moreover, we show that our delimitation approach can provide accurate inferences about the completion rate of speciation. As such, our results showcase not only the significant improvements that speciation-based delimitation can provide for inferences about species boundaries, but also the broad utility of the approach for studying the linkages of micro- and macroevolutionary processes. But importantly, because we model the biological reality of restrictions of gene flow before speciation (i.e., genetic structure within species), our study also explicitly addresses the fundamental limits of genetic-based delimitation, despite their popularity, or proclaimed superiority for systematic study (e.g., [40]). Namely, Multispecies Coalescent species delimitation inferences that rely on genetic data alone, without reference to any other information for delimiting species, are *not* reliable. Below we discuss the implications of our findings and what they suggest about the future of species delimitation, including possible directions for speciation-based delimitation in particular.

### Accurate to inaccurate inference

The substantial accuracy of species assignments when the identities of a subset of lineages are provided contrasts strongly with the relatively poor performance of analyses using genetic data alone (i.e., without supplemental information), speaking to the limits of knowledge possible with genomic data used in isolation. However, even so, while actual species identity assignments may remain challenging without supplemental information, even in these cases inferences regarding the number of species are remarkably robust generally [Fig 3a]. In particular,

these estimated species numbers are also markedly more reliable than those inferred under the MSC (see Fig 2a, [2], which shows the the MSC dramatically overestimates species numbers).

With respect to inferring the species status of unknown lineages, the delimitation model with the highest probability corresponded to the true partition under a broad parameter space, with two notable exceptions [Fig 3b]. First, accurately identifying the species status of all lineages is unlikely if half of those lineages have no information about them to constrain the inference procedure. Under such situations, the only property that can be reliably estimated is the number of species [Fig 3a]. Second, the probability of inferring the correct delimitation model depends upon the history of diversification itself. In particular, it is unlikely to correctly identify the species status of all lineages when the speciation-completion rate is very high. However, as we emphasized earlier, we simulated data over a parameter space designed to identify the theoretical limits, not just the promise, of DELINEATE. As such, the reported poor performance in specific areas of parameter space does not necessarily imply a limited utility of speciation-based delimitation in practice.

In practice, with more biologically realistic speciation-completion rates (e.g., half or tenth of the population isolation rates, as opposed to the relatively unrealistic cases where populations form at the same rate as species), and with even a little information in the form of known species assignments of some population lineages, the performance of DELINEATE increases dramatically. For example, accuracy approaches the 80%—90% range of simulated datasets in which the species status of all unknown lineages were correctly inferred. Note that in DELINEATE the user does not need to provide any information regarding the speciation process itself in the form of the speciation-completion rate parameter ($\sigma$): DELINEATE "learns" this from analysis of the data based on the known species assignments. Moreover, despite noise in the estimation of this parameter [Fig 4], estimation of the actual species delimitation model seems to still perform relatively well as long as the true rates themselves are not extreme (i.e., speciation-completion rates approaching parity with population isolation rates) [Fig 3].

Irrespective of whether the focus is on delimiting all, or just a subset, of lineages with unknown species status (or on speciation dynamics rather than delimitation *per se*; as discussed below), the study design under our new approach will differ from those in the past. In particular, investigators should adopt study designs that include lineages of known species identities, in addition to the lineages that they wish to assign to species, when they collect genetic data. That is, instead of restricting analysis to a set of genomic data collected in individuals in which we have no idea as to any of the species assignments, systematic studies should design analysis to span a broader context that includes at least some lineages of known species identities. Many species delimitation studies in fact do this routinely, as it is rather unusual for a system not to have *any* information about species identities for *any* of its lineages. This study design parallels those for analyses of divergence times in which the operational taxonomic units (i.e., the tips of the tree) are selected to include taxa for which calibration or fossil data is available. In the context of DELINEATE, the relationship of the number of species identities known *a priori* to accuracy is the simplest to understand, at least on a trivial level: the more information that we provide to the model, the better the model performs. In addition, it should be noted that the benefits of this information are not only in terms of informing the model, but also restricting parameter space in terms of the number of partitions to visit, thus speeding up computational times. However, computational time also becomes an important component the higher the total number of lineages. Although the amount of information about the speciation process that can be gleaned increases under such conditions, and allows for better inference about the delimitation model (see [Figs 3 and 4]), there is a computational trade-off. With larger datasets, the accuracy of inferences improves, but the number of partitions to be scored grows very quickly, making calculations infeasible when analyzing too many lineages.

## Speciation dynamics

We note that the incorporation of an explicit speciation process opens new frontiers not only in species delimitation analysis, but also in macroevolutionary studies of diversity. Specifically, and in particular using the model applied in DELINEATE, the rate of development of species isolation mechanisms, as distinct from the rate of population isolation, can be directly estimated. This is a valuable evolutionary biology study objective in its own right [41]. But, in addition, this provides investigators with a framework for studying the linkages between population and species-level processes. For example, understanding why species diversity differs among geographic areas or among taxa requires an understanding of how diversity is generated and maintained. As such, speciation-based delimitation approaches like DELINEATE can be used to address such questions, including testing "museum" vs "cradle" models to explain the higher diversity in the tropics compared with temperate areas (i.e., the "museum" with lower extinction rates or the "cradle" with higher speciation rates in the tropics) [42–53]. Instead of just characterizing "cradle" areas as having higher speciation rates, with estimates of the speciation-completion rate, we can ask whether the higher diversity reflects higher rates of population isolation (perhaps due to complex or dynamic geographies) or higher rates of development of speciation isolation mechanisms (see [53]), because these two processes that affect the duration of speciation [29] are decoupled in our model. Just as importantly, evolutionary biologists [53–62] have long highlighted the need and importance for modeling speciation as an extended process as done by the PBD, and the modeling of lineage splitting (population isolation) and species development as two separate processes in the PBD has been shown to provide novel and important insight into understanding how diversity is generated and maintained [26, 29, 53]. The nuances and ramifications of these two different paths to higher speciation rates provide deeper insight into the evolutionary history of a system by building a better understanding of how patterns at evolutionary time-scales are shaped by mechanisms and processes at ecological time-scales [53, 61]. Distinguishing between high rates of population isolation versus development of speciation isolation mechanisms are also useful for analyzing some interesting modes of speciation, such as ephemeral or ecological speciation [59, 62]. Insights about the speciation-completion rate as estimated by DELINEATE may explain how macroevolutionary patterns are regulated by microevolutionary processes.

## "Objective" species delimitation

DELINEATE allows for existing taxonomic knowledge, subjective or otherwise, to be incorporated into an objective species delimitation analysis. In contrast, when using the MSC alone for species delimitation, species boundaries are inferred algorithmically entirely from genomic data, without requiring any pre-existing taxonomic information. This might lead to the perception that the MSC is an entirely objective analysis in comparison with DELINEATE, as the MSC does not require or make use of any subjective information with regard to species status, concepts, or criteria. However, this characterization of the MSC is misleading.

The MSC adopts a single criteria for delimiting species boundaries: any and all detectable restrictions of gene flow. The criteria is subjective in the sense that it was not selected through an objective statistical optimization procedure, nor does it represent a scientific consensus regarding species boundaries that is universally accepted by all investigators for all systems. Furthermore, as it is a necessary assumption made when using the classical MSC alone for species delimitation, it remains an implicit subjective choice even if it was not explicitly stated, understood, or put forward by the investigator. Thus, while the classical MSC model does indeed provide an objective approach to species delimitation, it does so under a specific subjective species criteria or concept, albeit perhaps one not always recognized by investigators.

This subjective species boundary criteria might be valid for some systems. However, it is clearly invalid in many systems in nature—that is, in systems with multiple within-species population lineages (e.g., any species with population structure [10, 18, 20]), even if divergence occurs with gene flow (i.e., divergence with gene flow models based on the MSC, such as [63], are insufficient and will be misleading when applied to any system in nature in which there is detectable within-species population genetic structure).

Like the MSC, DELINEATE, too, provides for an objective species delimitation analysis under subjective criteria. However, unlike the MSC, this subjective criteria is not fixed and forced upon the study regardless of whether it is valid or not. Instead, DELINEATE allows for the criteria to vary based on the investigator's particular understanding of what constitutes a species.

## Distinguishing between species and population boundaries by modeling the speciation process

By conducting analyses that rely only on genetic data, with no other information to inform species delimitation (i.e., the unconstrained mode of inference in DELINEATE), our study speaks to the limits of knowledge and how much we can learn from genomic data alone. That is, with no information at all as to the species identities of any populations, while the true species partition is found within the 95% confidence interval, it is the best-preferred delimitation in only 8% of the cases. This is not a novel finding: such a non-integrated approach to species delimitation analysis—where a set of genomic data is used with no supporting or corroborating information and an algorithm is expected to "magically" [23] distinguish between populations and species boundaries to diagnose species—has always been problematical and unreliable [2–17, 20, 21].

Because we provide the true speciation-completion rate, estimation of the parameter itself is not compromising our model's performance. As such, our work shows that the inherent limitation arises from distinguishing genetic structure associated with populations versus species. That is, the actual challenge for accurately delimiting species (as well as what makes the MSC an inadequate model for species delimitation) is the presence of *restrictions in gene flow before speciation* rather than gene flow after speciation. Yet, this issue has received very little attention (at least in theoretical treatments). Instead, a popular focus has been on gene flow after speciation (e.g., [64]), as if it is also the central problem with applications of the MSC for species delimitation. Hopefully this study will help dispel this misconception and future work can focus on how methods might provide robust inference by contending with genetic structure that arises before speciation. Genetic structure within populations before speciation is the fundamental impediment to more general genetic-based applications (e.g., [2–4, 9, 11, 20]), as well as for the new class of speciation-based delimitation models we introduce here.

We acknowledge that there are a number of limitations and simplifying assumptions with our approach as currently implemented in DELINEATE– e.g., we assume a constant fixed speciation-completion rate, and we face computational challenges with large numbers of lineages with unknown species identities. We found, for example, that analyses with more than 15 population lineages with unknown identities were the limit that could be executed without recourse to machines of 1TB or more of memory. Note that this number, 15, is specifically the number of population lineages with *unknown* species assignments; the entire analysis could easily consist of several hundred or more population lineages as long as most of these were of known species assignments. Given that the principal computational challenge in our current implementation is the requirement to enumerate all possible partitions, adopting any of the standard optimization heuristics such as hill-climbing for maximum likelihood estimation or

various forms of MCMC for Bayesian estimation in future work should increase the efficiency of DELINEATE. With this increased efficiency, analysis of larger datasets are possible, and with the higher information content of these larger datasets, we are optimistic that the efficacy of DELINEATE will increase as well. This potentially provides an opening for more sophisticated modeling to capture the biological realities of diversification dynamics, such as differing speciation-completion rates across taxa.

Nevertheless, even with the current limitations, the big picture that emerges is this: the accuracy of species delimitation is improved with modeling of the speciation process. This modeling not only allows us to avoid conflating genetic structure within species with that between species [Fig 3a], it also allows us to ask and answer more sophisticated questions in macroevolutionary biology (see [Fig 4]). There are many different speciation processes that can be considered that will prove useful in this regard. For example, Morlon *et al.* [65] described 13 theoretical models and 8 empirical patterns of speciation. Our adoption of the protracted speciation model [26, 27, 29, 66] is, in fact, just one of this variety. We both hope and expect that other speciation models that better reflect either the realities of particular biological systems, or the perspectives of other investigators, will be incorporated into speciation-based delimitation approaches in the future.

## Supporting information

**S1 Text. A description of the general DELINEATE dynamic programming algorith is provided in S1 Text.**
(PDF)

**S1 Data. All scripts required to replicate our analyses are provided in S1 Data.** This is an compressed archive that includes: scripts to simulate data, construct analysis pipelines, and cluster execution job files (delineate-performance-setup/bin); some notes on the parameter space we used (delineate-performance-setup/docs); scripts to collate, compile, and analyze results, as well as generate plots/figures from results data (delineate-performance-results/bin/); CSV/TSV files summarizing each replicate, including (true) parameters as well as inferred parameter values and probablities, as well as metadata such as analysis execution date/time, cluster location, etc. (delineate-performance-results/data/extracts); TSV files containing data simulation/generation logs, including random seeds etc. (delineate-performance-results/data/logs). Note that we omit the full (simulated) data sets due to size (> 5TB). However, these can be easily regenerated in identical detail using the same random seeds for the data generation (given in the "logs") with the scripts found in the "setup" section above. Note also that the automatically generated logs provided above span a broad variety of studies and analyses, including not only the production runs reported here but also pilot runs, experimental studies, etc. Production run details relevant to this paper can be identified by correlating date/time/cluster with the information found in the "extracts" subdirectory above.
(TBZ)

## Author Contributions

**Conceptualization:** Jeet Sukumaran, Mark T. Holder, L. Lacey Knowles.

**Data curation:** Jeet Sukumaran, Mark T. Holder, L. Lacey Knowles.

**Formal analysis:** Jeet Sukumaran, Mark T. Holder, L. Lacey Knowles.

**Funding acquisition:** Jeet Sukumaran, Mark T. Holder, L. Lacey Knowles.

**Investigation:** Jeet Sukumaran, Mark T. Holder, L. Lacey Knowles.

**Methodology:** Jeet Sukumaran, Mark T. Holder, L. Lacey Knowles.

**Project administration:** Jeet Sukumaran, Mark T. Holder, L. Lacey Knowles.

**Resources:** Jeet Sukumaran, Mark T. Holder, L. Lacey Knowles.

**Software:** Jeet Sukumaran, Mark T. Holder, L. Lacey Knowles.

**Supervision:** Jeet Sukumaran, Mark T. Holder, L. Lacey Knowles.

**Validation:** Jeet Sukumaran, Mark T. Holder, L. Lacey Knowles.

**Visualization:** Jeet Sukumaran, Mark T. Holder, L. Lacey Knowles.

**Writing – original draft:** Jeet Sukumaran, Mark T. Holder, L. Lacey Knowles.

**Writing – review & editing:** Jeet Sukumaran, Mark T. Holder, L. Lacey Knowles.

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
