## [Decision Letter · Decision Letter 0]

1 Oct 2020

Dear Dr. Sukumaran,

Thank you very much for submitting your manuscript "Incorporating the Speciation Process into Species Delimitation" for consideration at PLOS Computational Biology.

As with all papers reviewed by the journal, your manuscript was reviewed by members of the editorial board and by several independent reviewers. In light of the reviews (below this email), we would like to invite the resubmission of a significantly-revised version that takes into account the reviewers' comments.

I share the opinions provided by both reviewers that this manuscript describes an interesting approach that is in need of some major revision to make it suitable for publication. I would expect a revision to address their comments critically with substantive changes. A few comments of my own to add:

1) The reviewers make an important point that it is not fully apparent to superficial readers of this manuscript that the model can only delimit species if some species are already defined. I think this should be clearly explained up front, returned to later, and apparent from the abstract.

2) Line 17-20. The fact that populations will be artificially promoted by MSC methods is independent of species concept, but what represents the correct level to define a species is very much dependent on species concept. A major absence in the introduction is defining what you mean by species – without that definition, it cannot be possible to objectively delimit them with a statistical analysis. I think you define them as “what systematists define them to be”, i.e. that information external from the method is used to define enough species that you can estimate your parameters. I don’t have a problem with that as a definition but it needs to be clearly stated. Without it, species have some mythical status in this approach and appear to be conjured from thin air, as nothing in the model actually specifies the correct level unless you know the parameter – which you can't do without knowing some species already! There is some useful discussion later on, but it needs to be much clearer upfront.

We cannot make any decision about publication until we have seen the revised manuscript and your response to the reviewers' comments. Your revised manuscript is also likely to be sent to reviewers for further evaluation.

Sincerely,

Timothy G. Barraclough

Guest Editor

PLOS Computational Biology

Stefano Allesina

Deputy Editor

PLOS Computational Biology

I share the opinions provided by both reviewers that this manuscript describes an interesting approach that is in need of some major revision to make it suitable for publication. I would expect a revision to address their comments critically with substantive changes. A few comments of my own to add:

1) The reviewers make an important point that it is not fully apparent to superficial readers of this manuscript that the model can only delimit species if some species are already defined. I think this should be clearly explained up front, returned to later, and apparent from the abstract.

2) Line 17-20. The fact that populations will be artificially promoted by MSC methods is independent of species concept, but what represents the correct level to define a species is very much dependent on species concept. A major absence in the introduction is defining what you mean by species – without that definition, it cannot be possible to objectively delimit them with a statistical analysis. I think you define them as “what systematists define them to be”, i.e. that information external from the method is used to define enough species that you can estimate your parameters. I don’t have a problem with that as a definition but it needs to be clearly stated. Without it, species have some mythical status in this approach and appear to be conjured from thin air, as nothing in the model actually specifies the correct level unless you know the parameter – which you can't do without knowing some species already! There is some useful discussion later on, but it needs to be much clearer upfront.

Reviewer's Responses to Questions

**Comments to the Authors:**

Reviewer #1: This manuscript describes a novel method for species delimitation using the protracted speciation model, which explicitly models splits of populations within species and their subsequent transitions to full species. Authors claim that they can correctly distinguish species from populations and can avoid over-splits of species by including the population-species transition processes.

I think incorporating the model of speciation process is a logical idea to correct the over-splits caused by the MSC-based delimitation. Also, this method can be useful not just for delimitation but for studies of speciation process itself. Nevertheless, I do not think that some of the authors claims in the manuscript is fully convincing and some corrections are required before publication. Especially, that the genomic sequences alone cannot correctly delimit species is not supported by their results. I think it is rather due to their specific model design.

As already pointed out in the Leaché et al. 2018 paper, the protracted speciation (PS) model decouples speciation process and lineage splitting process. The PS model maps transition events on a population tree with a constant rate (sigma). This parameter is independent from the lineage branching process and is not directly affected by population processes like genetic differentiation or gene flow. Therefore, the inference of sigma is impossible with tree branch lengths alone. The species assignment of some tips is always required to infer the transition rate under this model setting. The manuscript should more clearly mention this property.

In real situations, speciation process is very likely to be affected by multiple population level processes, and the transition rate likely correlated with these parameters (Such as Ne*m). Sigma probably could be estimated from them. Without testing the adequacy of the constant-independent-rate PS model, it is not safe to conclude that the delimitation is impossible from genomic data alone.

Apart from this point, I think the explicit modeling of speciation process is a promising approach and guiding delimitation with taxonomic information is still useful in many situations.

Other points:

Line 51.

The gdi thresholds in Jackson et al. and Leaché et al. are informed by biological observations. They set it to match with the observed species-population boundaries. This procedure is similar to estimating the transition rate from taxonomic knowledge used in this manuscript.

Line 79.

A brief model description is required here before reporting results.

Line 92.

"Figure 1" is missing in the text.

Reviewer #2: This manuscript describes use of the protracted birth-death (PBD) model for the purpose of species delimitation, as implemented in the DELINEATE software. A motivation for the work is to avoid the oversplitting that is commonly observed with species delimitation methods based on the multispecies coalescent, which is facilitated by the speciation-completion rate parameter in the PBD model. The application of this model to the problem of species delimitation is interesting and timely, and the DELINEATE software is nice addition to the collection to the set of tools available for this problem.

However, there are a few major issues with the manuscript that must be corrected before publication:

(1) The entire model is not formally described anywhere in the manuscript, making it impossible for the reader to fully understand how the software works. The PBD model is described to some extent in Figure 1, but Figure 1 is not referenced anywhere in the manuscript. However, the parameter $\\sigma$ doesn’t appear in the caption to Figure 1, although speciation completion is discussed there, and this parameter appears to be the key component of the model for this purpose, as it is discussed throughout the text. It seemed that perhaps the model would be carefully described in the section labeled “Statistical Model Description and Inference Algorithm”, but it is not. It is stated that the data are sequences, and partitions of sequences into species are defined. But how, specifically, are the sequence data integrated with the PBD model? How is $Pr(\\Lambda | \\sigma, S)$ actually computed? It is imperative that the full model be carefully defined somewhere in the manuscript.

The authors also need to more carefully define the algorithm used in DELINEATE and/or specify what options were used in running the simulations. My guess is that all partitions are attempted, the likelihood is calculated on a fixed lineage tree for each partition, and the one with the highest probability is selected. If this is indeed correct, it should be stated. Also, how is a 95% confidence set obtained? Are partitions added in decreasing order of probability until 95% of the probability is accounted for? Or something else?

(2) Is the species completion rate underestimated in these types of models because some species die out? It might be good to comment on this.

(3) For the unconstrained simulations, is there any pattern in the kinds of partitions that had the highest probability? Does the method tend to underestimate or overestimate the number of species? Are certain scenarios less likely to be detected than others?

(4) The first sentence of the Discussion doesn’t seem to be to be supported directly by any of the analyses. Is there evidence presented that this method prevents oversplitting, or is this comment just based on the fact that the model is expected to do that?

Similarly, while I don’t necessarily disagree with the claim (lines 152-153) that “.. inferences that rely on genetic data alone, without reference to any other information for delimiting species, are not reliable”, I don’t see that this claim is justified by the work presented here. Just because this model doesn’t allow for accurate delimitation doesn’t mean that there is no model for genetic data that can’t produce an accurate delimitation.

Minor comments/typos:

- line 27, “for their particular” — complete the phrase

- lines 35-39, problem with the latex citations

- line 44, maybe remove “the” before “MSC-based”

- line 58, remove the comma after “divergence”

- line 141, “under at reasonable” — re-word

- line 157, combine the first two paragraphs of this section

- line 214, missing punctuation

- line 255, “any” is repeated

- line 264, “as it is necessary assumption” — re-word

- first paragraph of Materials & Methods reads awkwardly — might be better to say something like “DELINEATE” has 3 modes of inference: …”

- Section on “Constrained Species Delimitation”, this is very difficult to read since the model has not yet been introduced. What does the parameter $\\sigma$ mean?

- line 394, extra semi-colon

- paragraph starting on line 383, I found this redundant/unnecessary

-line 413, refers to the PBD model, not yet introduced

- line 444, “speciation” is misspelled

- line 451, “the” is repeated

- line 505, “partition” -> “partitions”

- Caption to Fig. 3, “minimum” is misspelled

- When referencing the journal Systematic Biology, capitalize both words.

**Have all data underlying the figures and results presented in the manuscript been provided?**

Reviewer #1: Yes

Reviewer #2: Yes

PLOS authors have the option to publish the peer review history of their article (what does this mean?). If published, this will include your full peer review and any attached files.

Reviewer #1: No

Reviewer #2: No
---

## [Decision Letter · Decision Letter 1]

8 Mar 2021

Dear Dr. Sukumaran,

Thank you very much for submitting your manuscript "Incorporating the Speciation Process into Species Delimitation" for consideration at PLOS Computational Biology. As with all papers reviewed by the journal, your manuscript was reviewed by members of the editorial board and by several independent reviewers. The reviewers appreciated the attention to an important topic. Based on the reviews, we are likely to accept this manuscript for publication, providing that you modify the manuscript according to the review recommendations.

The manuscript now presents an improved account of the method and its fit into the context of other methods. I agree with all the comments by reviewer 1, so please can you complete minor revisions to address the remaining questions. I agree that the manuscript would be much clearer with Methods first, so please do this now.

Sincerely,

Timothy G. Barraclough

Guest Editor

PLOS Computational Biology

Stefano Allesina

Deputy Editor

PLOS Computational Biology

[LINK]

The manuscript now presents an improved account of the method and its fit into the context of other methods. I agree with all the comments by reviewer 1, so please can you complete minor revisions to address the remaining questions. I agree that the manuscript would be much clearer with Methods first, so please do this now.

Reviewer's Responses to Questions

**Comments to the Authors:**

Reviewer #1: This is a revised version of the manuscript describing DELINEATE, a species delimitation method which explicitly models species-population transition processes.

The authors did not agree with some of my comments in the last review, but at least they made some essential corrections. I think the manuscript now more correctly delivers their findings than the last version, and I agree that the MSC (but not genetic species delimitation in general) can be unreliable and external information helps improve delimitation accuracy. Although I am not fully convinced by their response, especially about the interpretation of gdi, I leave them to readers' decisions. I have some comments on the descriptions of the model, and I wrote them in the “Other comments”.

Also, I would recommend moving the Materials&Methods before the Results. In the current version of manuscript, it is very hard to understand the Results section because no information about the modes of software and simulation setups is provided beforehand. (I asked them to add some sentences at the beginning of the Results, but they did not.) I think moving the M&M to the front is easier and PLOS computation biology's style looks flexible according to their guideline.

Other comments:

Line 454:

Providing an equation of Pr(Lambda|sigma, S) will help readers to understand this method. If I understand it correctly, its actual form is Pr(No. transition events>0 | sigma, branch length) or Pr(No. transition events==0 | sigma, branch length) assuming Poisson-distributed events and branch lengths are determined by S and Lambda. The calculation of an overall likelihood and dynamic programming can be clearly understood with an explicit equation too.

Line 466:

It is a bit confusing to use “likelihood” and “probability” interchangeably. I think that Pr(Lambda|sigma, S) is a likelihood function of partition, Lambda, and the algorithm calculate a likelihood of delimitation/partition.

**Have all data underlying the figures and results presented in the manuscript been provided?**

Reviewer #1: Yes

PLOS authors have the option to publish the peer review history of their article (what does this mean?). If published, this will include your full peer review and any attached files.

Reviewer #1: No

Figure Files:

Data Requirements:

Reproducibility:

References:

---

## [Editor Report · Decision Letter 2]

29 Mar 2021

Dear Dr. Sukumaran,

We are pleased to inform you that your manuscript 'Incorporating the Speciation Process into Species Delimitation' has been provisionally accepted for publication in PLOS Computational Biology.

Best regards,

Timothy G. Barraclough

Guest Editor

PLOS Computational Biology

Stefano Allesina

Deputy Editor

PLOS Computational Biology

---

## [Editor Report · Acceptance letter]

16 Apr 2021

PCOMPBIOL-D-20-01275R2 

Incorporating the Speciation Process into Species Delimitation

Dear Dr Sukumaran,

I am pleased to inform you that your manuscript has been formally accepted for publication in PLOS Computational Biology. Your manuscript is now with our production department and you will be notified of the publication date in due course.

With kind regards,

Katalin Szabo
